# Integrative $R$-learner of heterogeneous treatment effects combining experimental and observational studies

**Lili Wu**                LWU9@NCSU.EDU  and  **Shu Yang**                SYANG24@NCSU.EDU
*Department of Statistics, North Carolina State University*

**Editors:** Bernhard Schölkopf, Caroline Uhler and Kun Zhang

## Abstract

The gold-standard approach to estimating heterogeneous treatment effects (HTEs) is randomized controlled trials (RCTs) or controlled experimental studies, where treatment randomization mitigates confounding biases. However, experimental data are usually small in sample size and limited in subjects' diversity due to expensive costs. On the other hand, large observational studies (OSs) are becoming increasingly popular and accessible. However, OSs might be subject to hidden confounding whose existence is not testable. We develop an integrative $R$-learner for the HTE and confounding function by leveraging experimental data for identification and observational data for boosting efficiency. We form a regularized loss function for the HTE and confounding function that bears the Neyman orthogonality property, allowing flexible models for the nuisance function estimation. The key novelty of the proposed integrative $R$-learner is to impose different regularization terms for the HTE and confounding function so that the possible smoothness or sparsity of the confounding function can be leveraged to improve HTE estimation. Our integrative $R$-learner has two benefits: first, it provides a general framework that can accommodate various HTE models for loss minimization; second, without any prior knowledge of hidden confounding in the OS, the proposed integrative $R$-learner is consistent and asymptotically at least as efficient as the estimator using only the RCT. The experiments based on extensive simulation and a real-data application adapted from an educational experiment show that the proposed integrative $R$-learner outperforms alternative approaches.

**Keywords:** Causal inference; Double penalization; Empirical risk minimization; Hidden confounding; Series estimator

## 1. Introduction

Heterogeneous treatment effects (HTEs) are the causal effects of a treatment or an intervention on an outcome of interest given the subjects' characteristics, which is the key query in various areas, such as precision medicine (Hamburg and Collins, 2010), offline contextual bandit evaluation (Dudík et al., 2011) and personalized policy recommendations (Athey, 2017). Randomized controlled trials (RCTs) are the gold-standard method for evaluating the HTEs because the randomization of treatment allocation in an RCT ensures that treatment and control groups are comparable and biases are minimized to the extent possible. However, RCTs might be limited in sample size, due to the possible expensive costs and substantive efforts during the data collection procedure. Besides, RCTs might also be limited in subjects' diversity, due to the restrictive inclusion/exclusion criteria for enrollment. The limitations in both scale and scope of RCTs can compromise the causal effect estimation which leads to under-powered HTE estimators.

On the other hand, due to the larger sample size, broader diversity, and more accessibility, observational studies (OSs) have gained popularity. There is considerable interest in the literature to use OSs to improve the generalizability of the RCT findings (Dong et al., 2020; Lee et al., 2021; Wu and Yang, 2021). Besides, OSs have also been widely used for HTE estimation in various

domains such as healthcare and education. For example, clinicians can use electronic health records and disease registries to help recommend the individual-level treatment; teachers use school records to assign kids to different classes. However, due to lack of randomization, the OS has a universal concern of unmeasured confounders, which cannot be verified in practice with itself alone (Pearl, 2009). The no unmeasured confounding assumption requires all relevant predictors of treatment and outcome to be measured, which is barely held in the OS. For example, physicians prescribe a medicine based on the patient's conditions of which some are not recorded in medical charts. With the OS alone, the hidden confounding makes the HTE unidentifiable and can lead to a biased HTE estimation. To mitigate confounding bias with the OS alone, classic approaches include instrumental variables (Angrist et al., 1996) and sensitivity analysis (Robins et al., 2000). However, instrumental variables, variables that must be related to the outcome only through the treatment, might not exist and highly relies on domain knowledge. The sensitivity analysis is used to evaluate the robustness of the estimators to no unmeasured confounding, but it could not identify the confounding function. The RCT and OS have their own strengths and weaknesses, therefore, we are motivated to develop an integrative learner by leveraging the RCT for identifying the HTE and the OS for boosting its efficiency.

*Related work*. Data integration is a broad concept in computer science, medicine, and social science. Bareinboim and Pearl (2016) investigate the identification of causal estimands in "data-fusion" problems under the probabilistic graphical model framework. Kallus et al. (2018) propose to use an RCT sample to correct the bias of the HTE estimated on an OS sample, assuming that the bias can be well approximated by a function with low complexities, e.g., a linear function of the covariates. They, however, do not study the efficiency guarantees of their integrative estimators. Yang et al. (2020b) propose a pre-test estimator of the HTE by testing the comparability of the RCT and OS samples and deciding whether or not to combine the RCT and OS samples for subsequent estimation. This procedure prevents the final estimator from possibly large biases in the OS sample, but it does not fully utilize all OS information to improve HTE estimation.

Under the no unmeasured confounding assumption in the OS, there have been many methods to estimate the HTE, ranging from the classic methods such as regression-based methods and (augmented) inverse probability weighting methods (Robins et al., 1994; Scharfstein et al., 1999) to some recent more flexible methods based on machine learning models such as neural network (Shalit et al., 2017), random forest (Wager and Athey, 2018), and boosting (Powers et al., 2018). More recently, motivated by Robinson's transformation (Robinson, 1988), Nie and Wager (2021) formalize a loss function to estimate the HTE which can be optimized by arbitrary loss-minimization procedures. The resulting solution is called $R$-learner, which bears the Neyman orthogonality property(Neyman, 1959) that the $R$-learner has a root-N rate of convergence with weaker conditions on nuisance function approximation. Therefore, the $R$-learner can incorporate a much broader class of flexible machine learning models to approximate nuisance functions. Carrying the benefits of the $R$-learner, we develop an integrative $R$-learner of the HTE combining RCT data and OS data where the OS might be subject to hidden confounding. The key step is to formulate a regularized loss function for the HTE and confounding function with the Neyman orthogonality. More importantly, we impose different regularization terms for the HTE and confounding function, so that the resulting integrative $R$-learner can capture the possible smoothness or sparsity of the confounding function to improve HTE estimation. The main benefit of our proposed integrative $R$-learner is two-fold: first, it can incorporate flexible models for nuisance functions as well as the HTE and confounding function; second, without any prior knowledge of hidden confounding in the OS, the integrative $R$-

learner is consistent for the true HTE and asymptotically at least as efficient as the estimator using the RCT alone.

The rest of the paper is organized as follows. We introduce the basic setup in Section 2, the proposed $R$-learner in Section 3, and its theoretical analysis in Section 4. We describe two alternative methods for data integration in Section 5 and conduct extensive experiments for comparison based on simulated and real data in Section 6. Finally, we conclude the paper with a discussion in Section 7.

## 2. Data sources and identification assumptions

We focus on a setting with a binary treatment variable $A \in \{0, 1\}$, let $X \in \mathcal{X} \subset \mathbb{R}^p$ be a vector of pre-treatment covariates, and let $Y \in \mathbb{R}$ be the outcome of interest. We use the potential outcomes framework (Neyman, 1923; Rubin, 1974) to define causal effects and let $Y(a)$ be the potential outcome had the subject taken treatment $a$, for $a = 0, 1$. Then the HTE can be characterized by $\tau(X) = \mathbb{E}\{Y(1) - Y(0) \mid X\}$.

Two data sources are accessible: the RCT with independent and identically distributed (i.i.d.) $n$ subjects $\{(X_i, A_i, Y_i, S_i)\}_{i \in \mathcal{I}_n}$ and the OS with i.i.d. $m$ subjects $\{(X_i, A_i, Y_i, S_i)\}_{i \in \mathcal{I}_m}$, where $\mathcal{I}_n$ and $\mathcal{I}_m$ are index sets of the RCT and the OS, respectively. Let $S_i$ be the binary indicator of the $i$th subject in the RCT: $S_i = 1$ if $i \in \mathcal{I}_n$ and 0 if $i \in \mathcal{I}_m$. We denote two nuisance functions: the propensity score $e(X, S) = P(A = 1 \mid X, S)$, and the conditional mean outcome $\mu(X, S) = \mathbb{E}(Y \mid X, S)$. Our goal is to estimate $\tau(X)$, where $X$ belongs to the support of $X$ in the RCT. To realize the possible efficiency gains from the OS, we assume the support of $X$ in the RCT is nested within or overlapping with that in the OS because $\tau(X)$ is not identifiable outside the support of $X$ in the RCT.

One of the fundamental challenges in causal inference is the identification of the HTE because $Y(1)$ and $Y(0)$ cannot be jointly observed on the same subject. Therefore, to overcome this challenge, we invoke the two common assumptions in the causal inference (e.g., Rubin, 1980; Rosenbaum and Rubin, 1983; Tsiatis, 2006) and data integration literature (e.g., Colnet et al., 2020; Degtiar and Rose, 2021).

**Assumption 1** *(Ignorability of treatment assignment in the RCT and causal consistency):* *(a)* $A \perp\!\!\!\perp \{Y(0), Y(1)\} \mid (X, S = 1)$. *(b) There exist* $c_1, c_2 \in (0, 1)$ *such that* $e(X, S) \in [c_1, c_2]$ *for all* $X \in \mathcal{X}$. *(c)* $Y = Y(A)$.

**Assumption 2** *(Transportability of the HTE):* $\mathbb{E}\{Y(1) - Y(0) \mid X, S = s\} = \tau(X), s = 0, 1$.

Assumption 1(a) is also called no unmeasured confounding, which holds in the RCT by default due to the randomized treatment assignment. Under Assumption 1, the HTE can be identified based on the RCT. On the contrary, we do not impose the ignorability of treatment assignment for the OS, which may be stringent in practice. Assumption 2 is needed to extend the definition and identification of causal effects from the RCT to the OS, which holds when $X$ captures all treatment effect modifiers. Its plausibility relies on experts' domain knowledge. It is a common assumption in the data integration literature and the weakest one among the mean exchangeability assumption (Dahabreh et al., 2019), i.e., $\mathbb{E}\{Y(a) \mid X, S = s\}$ and ignorability of study participation, i.e., $Y(a) \perp\!\!\!\perp S \mid X, (a = 0, 1)$ (Stuart et al., 2011; Buchanan et al., 2018).

## 3. The proposed integrative $R$-learner

Before delving into technical details, we provide a road map of the proposed framework: $(a)$ identification of the HTE and confounding function, $(b)$ formulation of the loss function, and $(c)$ the proposed two-step estimation procedure that tackles two difficulties associated with loss optimization. We will expand them in detail in the remaining of this section.

### 3.1. Identification of the HTE and confounding function and formulation of the loss functions

When lack of the ignorability of treatment assignment, we define the bias between the difference in conditional mean outcomes in the OS and the HTE as the confounding function

$$c(X) = \mathbb{E}(Y \mid X, A = 1, S = 0) - \mathbb{E}(Y \mid X, A = 0, S = 0) - \tau(X). \tag{1}$$

By Assumption 1$(c)$, $\mathbb{E}(Y \mid X, A = a, S = 0) = \mathbb{E}\{Y(a) \mid X, A = a, S = 0\}$. If all the pre-treatment confounding variables are captured in the OS, i.e., $A \perp\!\!\!\perp \{Y(0), Y(1)\} \mid (X, S = 0)$, then $\mathbb{E}\{Y(a) \mid X, A = a, S = 0\} = \mathbb{E}\{Y(a) \mid X, S = 0\}$, which implies $c(X) = 0$ under Assumption 2. The confounding function is not identified based on the OS alone, however, we show that combining the RCT and OS can identify the HTE and confounding function. The key step toward identifying and estimating the HTE and confounding function is introducing the residual

$$\epsilon = Y - \mathbb{E}(Y \mid X, A = 0, S) - A\{\tau(X) + (1 - S)c(X)\}. \tag{2}$$

Under Assumptions 1-2, we can show the fact $\mathbb{E}(\epsilon \mid X, A, S) = 0$ as follows:

a) Conditioning on $(X, A = 0, S)$: $\mathbb{E}(\epsilon \mid X, A = 0, S) = 0$;

b) Conditioning on $(X, A = 1, S = 0)$: $\mathbb{E}(\epsilon \mid X, A = 1, S = 0) = \mathbb{E}(Y \mid X, A = 1, S = 0) - \mathbb{E}(Y \mid X, A = 0, S = 0) - \{\tau(X) + c(X)\} \overset{(1)}{=} 0$;

c) Conditioning on $(X, A = 1, S = 1)$: $\mathbb{E}(\epsilon \mid X, A = 1, S = 1) = \mathbb{E}(Y \mid X, A = 1, S = 1) - \mathbb{E}(Y \mid X, A = 0, S = 1) - \tau(X) = \mathbb{E}\{Y(1) \mid X, S = 1\} - \mathbb{E}\{Y(0) \mid X, S = 1\} - \tau(X) = 0$. The last second equality is implied by Assumptions 1 and the last equality is implied by Assumption 2.

Besides, from the definition of $\mu(X, S)$, we have

$$\begin{aligned} \mu(X, S) &= \mathbb{E}(Y \mid X, A = 1, S)e(X, S) + \mathbb{E}(Y \mid X, A = 0, S)\{1 - e(X, S)\} \\ &= \mathbb{E}(Y \mid X, A = 0, S) + \{\tau(X) + (1 - S)c(X)\}e(X, S). \end{aligned} \tag{3}$$

Combining (2) and (3), we have

$$Y - \mu(X, S) = \{\tau(X) + (1 - S)c(X)\}\{A - e(X, S)\} + \epsilon, \tag{4}$$

which motivates the mean squared error loss function minimization problem for identifying $\{\tau(\cdot), c(\cdot)\}$

$$\{\tau(\cdot), c(\cdot)\} = \underset{\tilde{\tau}, \tilde{c}}{\operatorname{argmin}} \left( \mathbb{E}\left[Y - \mu(X, S) - \{\tilde{\tau}(X) + (1 - S)\tilde{c}(X)\}\{A - e(X, S)\}\right]^2 \right), \tag{5}$$

Naturally, (5) leads to the "empirical" loss function to estimate the HTE and confounding function:

$$\underset{\tilde{\tau}, \tilde{c}}{\operatorname{argmin}} \left( \mathbb{P}_N \left[Y_i - \mu(X_i, S_i) - \{\tilde{\tau}(X_i) + (1 - S_i)\tilde{c}(X_i)\}\{A_i - e(X_i, S_i)\}\right]^2 + \Lambda_\tau + \Lambda_c \right), \tag{6}$$

where $N = n + m$, $\mathbb{P}_N(X_i) = N^{-1} \sum_{i \in \mathcal{I}_n \bigcup \mathcal{I}_m} X_i$, and $\Lambda_\tau$ and $\Lambda_c$ are regularizers on the complexity of the $\tau(\cdot)$ and $c(\cdot)$ functions to avoid overfitting. The penalties could be added directly on the coefficients of the features such as penalized regression, or indirectly on the black-box models such as boosting (Mason et al., 1999) or random forest (Breiman, 2001) by restricting the depth of the decision trees.

It is worth discussing the relationship of the proposed estimator with existing approaches. If replacing the functions $\tau(X) + (1 - S)c(X)$ in (4) with finite-dimensional parameters, it coincides with the decomposition proposed in Robinson (1988) to estimate parametric components in partially linear models. This decomposition bears Neyman orthogonality (Neyman, 1959) to obtain $\sqrt{N}$-rate estimation of target parameters while the convergence rates of the nuisance functions are slower than $\sqrt{N}$, which has recently also been called "rate double robustness" (Rotnitzky et al., 2021). This approach has gained considerable attention in the causal inference community for treatment effect estimation such as G-estimation (Robins, 2004), double machine learning (Chernozhukov et al., 2018), causal forest (Athey et al., 2019) and $R$-learner (Nie and Wager, 2021). However, the aforementioned work only considers causal effect estimation in the data satisfying the no unmeasured confounding assumption. The proposed loss function can be viewed as an extension (although not straightforward) of the ones of Robinson (1988) and Nie and Wager (2021) to the data integration context combining the RCT and OS. Thus, we call the proposed method "integrative $R$-learner".

In order to solve (6), however, two difficulties arise: (i) Except that the propensity score in the RCT $e(X, S = 1)$ might be known, other nuisance functions $\mu(X, S)$ and $e(X, S = 0)$ are unknown; and (ii) $\tau(\cdot)$ and $c(\cdot)$ may have different complexities, but rare off-the-shelf software can directly solve (6) with different $\Lambda_\tau$ and $\Lambda_c$. We will show that the possible smoothness or sparsity of $c(\cdot)$ can be leveraged to improve the estimation of $\tau(\cdot)$ over using the RCT alone (Theorem 1). Thus, using different $\Lambda_\tau$ and $\Lambda_c$ is critical to actualize the potential gain from the OS for HTE estimation. In the next two subsections, we will describe how we address the two concerns.

### 3.2. Estimation of the nuisance functions

Cross-fitting is an increasingly popular prediction method to incorporate flexible machine learning approaches in classical semiparametrics (Schick, 1986; Robins et al., 2008, 2017; Chernozhukov et al., 2018; Newey and Robins, 2018; Kennedy, 2020; Nie and Wager, 2021). Not only does it help to make theories more elegant, but also to overcome the overfitting/high-complexity phenomena that commonly arise in highly adaptive machine learning methods (Chernozhukov et al., 2018). It is a simple procedure by splitting data into fitting and prediction parts.

We adopt cross-fitting to estimate the nuisance functions. We randomly split the data into $K$ (commonly set to be 5) equal-size folds. Each subject $i$ falls into one and only one data fold. Let $k(i)$ be the index set of the data fold where the subject $i$ belongs. We use the samples that do not belong to $k(i)$ to fit the user-specified models (such as lasso regression and Xgboost), which results in the estimated conditional mean outcome and propensity score models, denoted as $\hat{\mu}^{-k(i)}(\cdot, \cdot)$ and $\hat{e}^{-k(i)}(\cdot, \cdot)$ respectively. Next, using the estimated nuisance functions, we predict the function values on the subjects belonging to $k(i)$. Repeating the procedure on each fold, we finally obtain the nuisance function estimates of all the subjects, i.e., $\hat{\mu}^{-k(i)}(X_i, S_i)$ and $\hat{e}^{-k(i)}(X_i, S_i), i \in \mathcal{I}_n \cup \mathcal{I}_m$.

This step leads to the final empirical loss function for estimating the HTE and confounding function:

$$\{\hat{\tau}(\cdot), \hat{c}(\cdot)\} = \underset{\tilde{\tau}, \tilde{c}}{\operatorname{argmin}} \Bigg( \mathbb{P}_N \Big[ Y_i - \hat{\mu}^{-k(i)}(X_i, S_i) - $$
$$\{\tilde{\tau}(X_i) + (1 - S_i)\tilde{c}(X_i)\}\{A_i - \hat{e}^{-k(i)}(X_i, S_i)\} \Big]^2 + \Lambda_\tau + \Lambda_c \Bigg). \tag{7}$$

### 3.3. Series approximations of the HTE and confounding function

For the easy exposition, we now describe the integrative $R$-learner adopting series estimation of the HTE and confounding function, although the general framework can accommodate general flexible models. We will illustrate the generality of the proposed framework in the data application in Section 6.2. Series estimation is a widely applicable nonparametric method to approximate the unknown conditional expectation that can have a flexible functional form. The approximation consists of $D$ basis functions which have a broad class such as polynomial basis, wavelet basis, and spline, where $D$ grows with the sample size. Besides the versatility of the series estimators, their asymptotic properties have also been studied thoroughly in a large body of literature (see, e.g., Chen, 2007; Belloni et al., 2015, and the references therein).

We use the series method to approximate $\tau(X)$ and $c(X)$ as follows,

$$\tau(X) = p_\tau^\mathsf{T}(X)\beta_\tau + r_\tau, \ c(X) = p_c(X)^\mathsf{T}\beta_c + r_c, \tag{8}$$

where $p_\tau(X) = \{p_{\tau,1}(X), \ldots, p_{\tau,D}(X)\}^\mathsf{T}$ and $p_c(X) = \{p_{c,1}(X), \ldots, p_{c,D}(X)\}^\mathsf{T}$ are two vectors of basis functions, $D$ can increase with sample size $N$, and $r_\tau$ and $r_c$ are approximation errors. Combining (4) and (8) leads to the square loss function,

$$L(\beta) = \mathbb{P}[Y - \mu(X, S) - g(X, S; \beta)\{A - e(X, S)\}]^2,$$

where $g(X, S; \beta) = p_\tau^\mathsf{T}(X)\beta_\tau + (1 - S)p_c(X)^\mathsf{T}\beta_c$ and $\beta = (\beta_\tau^\mathsf{T}, \beta_c^\mathsf{T})^\mathsf{T}$. We denote the empirical loss function as

$$\widehat{L}_N(\beta) = \mathbb{P}_N \big[ Y_i - \hat{\mu}^{-k(i)}(X_i, S) - g(X_i, S_i; \beta)\{A - \hat{e}^{-k(i)}(X_i, S_i)\} \big]^2. \tag{9}$$

Since the number of basis functions controls the smoothness of series estimators, $\tau(X)$ and $c(X)$ may require a different number of basis functions due to their possibly different underlying complex nature. Thus, we require different regularization parameters. We will show that the possible smoothness or sparsity of $c(X)$ leads to the efficiency gain of the integrative $R$-learner compared to using the RCT alone. Thus, it is critical to impose different regularization parameters for $\tau(X)$ and $c(X)$. Specifically, let $\widehat{\beta} = \operatorname{argmin}_{b \in \mathbb{R}^{2D}} \big\{ \widehat{L}_N + \sum_{d=1}^{D} s_{\lambda_\tau}(|b_{\tau,d}|) + \sum_{d=1}^{D} s_{\lambda_c}(|b_{c,d}|) \big\}$, where $s_\lambda(\cdot)$ is the smoothly clipped absolute deviation (SCAD) penalty function (Fan and Li, 2001) and the tuning parameters $\lambda_\tau$ and $\lambda_c$ can be chosen through cross-validation by grid searching two prespecified ranges. One way to simplify the tuning process is to add a scale tuning parameter between $\lambda_\tau$ and $\lambda_c$. For example, let $sc = \lambda_\tau / \lambda_c$, and set the search range of the scale tuning parameter as $sc \in \{0, 0.5, 1, 1.5\}$, which represents that no penalty on $\tau(\cdot)$, the penalty on $\tau(\cdot)$ less than, equal to, and larger than that on $c(\cdot)$, respectively. For each $sc$, we can use cross-validation to choose $\lambda_c$ with the off-the-shelf software `cv.ncvreg` function in the R package `ncvreg` (Breheny and Huang,

2011). The final fitted model corresponds to the scale tuning parameter that has the minimum mean cross-validated error (see more details in Appendix A).

When the shrinkage parameters are complex and high-dimensional such as those in deep neural network, random forest, and Xgboost, we propose a variant of integrative $R$-learner by iteratively learning $\tau(\cdot)$ and $c(\cdot)$ (see Appendix D). The variant is illustrated in the real data experiment in Section 6.2.

## 4. Theoretical analysis

The main goal of our theoretical analysis is to show that our integrative $R$-learner is consistent and at least as efficient as the $R$-learner based only on the RCT. For concreteness, we focus on the $R$-learner based on series estimation. First of all, under the regularity conditions given in Fan and Li (2001), $\widehat{\beta}$ satisfies the selection consistency and oracle properties, i.e., $\|\widehat{\beta} - \beta\| = O_p(N^{-1/2})$. Thus, it suffices to focus on $\widehat{L}_N(\beta)$ in (9) for theoretical analysis. Replacing its estimated nuisance functions with the true ones, we denote it as $L_N(\beta) = \mathbb{P}_N\big[Y_i - \mu(X_i, S_i) - g(X_i, S_i; \beta)\{A_i - e(X_i, S_i)\}\big]^2$. The following two assumptions are used to assure the difference between $\widehat{L}_N(\beta)$ and $L_N(\beta)$ diminishes with a relatively fast rate with $N$ under which the Neyman orthogonality of the loss function renders the impact of the estimated nuisance functions negligible.

**Assumption 3** $\|\tau(X)\|_\infty$, $\|c(X)\|_\infty$ and $\mathbb{E}[\{Y - \mu(X, S)\}^2 \mid X, S]$ are bounded, for any $X \in \mathcal{X}, S \in \{0, 1\}$.

**Assumption 4** $\mathbb{E}\left[\{\mu(X, S) - \hat{\mu}(X, S)\}^2\right] = O(a_N^2)$ and $\mathbb{E}\left[\{e(X, S) - \hat{e}(X, S)\}^2\right] = O(a_N^2)$ for some sequence $a_N$ such that $a_N = O(N^{-r})$ with $r > 1/4$.

Assumption 3 is plausible in many practical problems. Assumption 4 relaxes the usual $\sqrt{N}$-consistency to a converge rate required to be only faster than $N^{-1/4}$, which can occur in a broad class of models such as single-index models, generalized additive models, partially linear models, and lasso regression (Horowitz, 2009; Belloni et al., 2014; Kennedy, 2016). Besides, we impose the standard assumptions for series estimation (Belloni et al., 2015). For simplicity, let $p_i = p(X_i, A_i, S_i) = \{A_i - e(X_i, A_i)\} \{p_\tau^\intercal(X_i), (1 - S_i)p_c(X_i)^\intercal\}^\intercal$, $\hat{p}_i = \{A_i - \hat{e}(X_i, A_i)\}\{p_\tau^\intercal(X_i), (1 - S_i)p_c(X_i)^\intercal\}^\intercal$, and $\xi_D = \sup_{x,a,s}\|p(x, a, s)\|$.

**Assumption 5** Uniformly over all $N$, eigenvalues of $\Gamma := \mathbb{E}(p_i p_i^\intercal)$ are bounded above and away from zero.

**Assumption 6** (a) For each $N$ and $D$, there are finite constants $c_D$ and $l_D$ such that for each $f \in \mathcal{F}$, $\|r_f\|_{F,2} := \sqrt{\int_{x \in \mathcal{X}} r_f^2(x)dF(x)} \leq c_D$ and $\|r_f\|_{F,\infty} := \sup_{x \in \mathcal{X}} |r_f(x)| \leq l_D r_D$. (b) $\sqrt{\xi_D^2 \log(D/N)}(1 + \sqrt{D}l_D c_D) \to 0$. (c) $l_D c_D \to 0$. (d) $\sqrt{N/D}l_D c_D \to 0$.

**Assumption 7** $\mathbb{E}(\epsilon^2 \mid X, S, A) = \sigma^2$ for some constant $\sigma$.

By the proposed method, the estimated treatment effect is $\hat{\tau}(X) = \hat{p}_\tau^\intercal \hat{\beta}_\tau$, where $\hat{\beta} := (\hat{\beta}_\tau^\intercal, \hat{\beta}_c^\intercal) = \operatorname{argmin}_b \mathbb{P}_N \left\{Y_i - \hat{\mu}^{-k(i)}(X_i, S_i) - \hat{p}_i^\intercal b\right\}^2$. If only using the RCT data, then the estimated treatment effect is $\hat{\tau}_{\mathrm{rct}} = \hat{p}_\tau^\intercal \hat{\beta}_{\mathrm{rct}}$, where $\hat{\beta}_{\mathrm{rct}} = \operatorname{argmin}_b \mathbb{P}_N S_i \left\{Y_i - \hat{\mu}^{-k(i)}(X_i, S_i) - \hat{p}_i^\intercal b\right\}^2$. Denote $\mathbb{V}(Z)$ as the asymptotic variance of the random variable $Z$ or the asymptotic covariance matrix of the random vector $Z$.

**Theorem 1** *Under Assumptions 1-7, $\hat{\tau}(x) - \tau(x) = O_p(N^{-1/2})$ for any $x \in \mathcal{X}$, and $\mathbb{V}(\hat{\tau}) \leq \mathbb{V}(\hat{\tau}_{rct})$. Moreover, the equality holds if and only if $p_\tau(X) = Mp_c(X)$, for some constant matrix $M$.*

Theorem 1 shows that the integrative $R$-learner is consistent and at least as efficient as the HTE estimator with only the RCT. More interestingly, when the underlying HTE $\tau(\cdot)$ is less restrictive than $c(\cdot)$, the integrative $R$-learner has strictly larger efficiency than the estimator with the RCT alone. Intuitively, we can start with one extreme example where there is no unmeasured confounding, which means that the $c(x) = 0$ (the simplest function). If we knew this fact, then we could combine the RCT and OS directly for HTE estimation, and the integrative estimator improves the efficiency of the RCT-only estimator obviously due to the larger sample size. Our proposed estimation method uses basis functions to approximate the confounding function and uses penalization to leverage the possible smoothness or sparsity of the confounding function to achieve the same result as the oracle one. We present a proof sketch here (the details are presented in Appendix B):

a) Under Assumptions 3-4, we can show: $\widehat{L}_N(\beta) = L_N(\beta) + O_P(a_N^2)$, where $a_N = O(N^{-r})$ with $r > 1/4$, which implies $\hat{\beta} = \operatorname{argmin}_b \mathbb{P}_N \left\{ Y_i - \mu(X_i, S_i) - p_i^\mathsf{T} b \right\}^2 + O_P(a_N^2)$.

b) By Pointwise Normality of series (see Theorem 4.2 in Belloni et al., 2015)

$$\sqrt{N}(\hat{\beta} - \beta) = \mathbb{E}(p_i p_i^\mathsf{T})^{-1} \sqrt{N}(\mathbb{P}_N - \mathbb{P}) \begin{pmatrix} p_\tau \epsilon \\ p_c \epsilon \end{pmatrix} + o_P(1) + O_P\left( \sqrt{N} a_N^2 \right).$$

c) By algebra, we can show $\mathbb{V}^{-1}(\hat{\beta}_\tau) - \mathbb{V}^{-1}(\hat{\beta}_{\mathrm{rct}})$ is non-negative definite.

## 5. Other methods

We describe two alternative integrative HTE estimators. They both start with constructing an adjusted outcome $\widetilde{Y}$ such that $\mathbb{E}(\widetilde{Y} \mid X, S = 1) = \tau(X)$ under Assumptions 1-2. A common choice of the adjusted outcome is

$$\widetilde{Y} = \frac{A\{Y - Q_S(X,1)\}}{e(X,S)} + Q_S(X,1) - \frac{(1-A)\{Y - Q_S(X,0)\}}{1 - e(X,S)} - Q_S(X,0), \tag{10}$$

where $Q_S(X, a) = \mathbb{E}(Y \mid X, A = a, S)$ (Huang and Yang, 2022). Then, one can fit $\widetilde{Y}$ on $X$ to obtain the HTE estimator. This approach is also a variant of the $U$-learner (Nie and Wager, 2021). In the following, the first integrative estimator constructs the similar outcome-adjusted equality for $c(X)$ in the OS, while the second one proposed by Kallus et al. (2018) estimates $c(X)$ based on imposing the linear model assumption and using the $\widetilde{Y}_i$ in the RCT as the unbiased estimates of $\tau(X_i)$.

### 5.1. Outcome-adjusted method

We show that the adjusted outcome for the OS can be used to identify $c(X)$ once $\tau(X)$ is identified from the RCT as stated in Proposition 2 (see the proof in Appendix C).

**Proposition 2** *Under Assumptions 1-2, we have*

$$\mathbb{E}(\widetilde{Y} \mid X, S = 0) = \tau(X) + c(X), \tag{11}$$

*where $\widetilde{Y}$ is (10).*

Here is one way to get the integrative estimator for the HTE by combining the facts $\mathbb{E}(\widetilde{Y} \mid X, S = 1) = \tau(X)$ and (11). We first fit $\widetilde{Y}$ on $X$ using the RCT to find important features, and then fit $\widetilde{Y}$ on $X$ again but using all the RCT and the OS samples with incorporating the selected features in $\tau(X)$ and all the features in $c(X)$. The procedure is summarized below.

a) Estimate the adjusted outcomes, denoted as $\widetilde{Y}_i$ , $i \in \mathcal{I}_n \cup \mathcal{I}_m$;

b) Fit the adjusted outcomes on basis functions $b_{Kn}(X)$ with lasso regression using the RCT samples: $\widetilde{Y}_i \sim \alpha^{\mathsf{T}} b_{Kn}(X_i)$, $i \in \mathcal{I}_n$; obtain the selected basis functions denoted as $b^*_{Kn}(X_i)$;

c) Lasso regression with the RCT and the OS samples on the selected basis functions $b^*_{Kn}(X_i)$ and basis functions $b_{Lm}(X_i)$ of the OS: $Y_i \sim \alpha^{\mathsf{T}} b^*_{Kn}(X_i) + \beta^{\mathsf{T}} b_{Lm}(X_i)$. Note that $b_{Lm}(X_i) = 0, i \in \mathcal{I}_n$, and the penalty is only added on $\beta$;

d) Obtain the estimated HTE $\hat{\tau}(X) = \hat{\alpha}^{\mathsf{T}} b^*_{Kn}(X)$ , where $\hat{\alpha}$ is got from step c).

However, this approach relies on the estimation of the adjusted outcomes and can be highly unstable since the inverse of the estimated propensity scores might result in extreme values. When there is only the RCT or OS data, one only requires Steps a) and b) for HTE estimation by fitting $\widetilde{Y}$ on $X$. This procedure provides unbiased estimates in the RCT due to $\mathbb{E}(\widetilde{Y} \mid X, S = 1) = \tau(X)$ but biased estimates in the OS due to the possible unmeasured confounders.

### 5.2. 2-step procedure proposed by Kallus et al. (2018)

Assuming a linear model for the confounding function $c(X) = \theta^{\mathsf{T}} X$, Kallus et al. (2018) propose an integrative 2-step HTE estimation procedure summarized below.

a) Use any regression methods, e.g., random forest or causal forest (Athey et al., 2019), to estimate $Q_0(X, 1)$ and $Q_0(X, 0)$ with the OS samples, then take its difference, denoted as $\hat{\omega}(\cdot) = \widehat{Q}_0(\cdot, 1) - \widehat{Q}_0(\cdot, 0)$;

b) Apply the definition in (1) to learn $c(X)$: $\hat{\theta} = \operatorname{argmin}_\theta \sum_{i \in \mathcal{I}_n} \left\{ \widetilde{Y}_i - \hat{\omega}(X_i) + \theta^{\mathsf{T}} X_i \right\}^2$, which gives the estimated HTE immediately $\hat{\tau}(x) = \hat{\omega}(x) - \hat{\theta}^{\mathsf{T}} x$.

This approach has a different loss function from ours and it shows the consistency of $\hat{\tau}(\cdot)$ at a rate that is governed by the rate of $\hat{w}(\cdot)$, but it has restrictions on the form of $c(X)$.

## 6. Experiments

We evaluate the finite-sample performances of our proposed method through simulated data and a real data application of the Tennessee STAR study which aims to measure the effect of class size on test scores (Word et al., 1990; Krueger, 1999). To illustrate the generality of our proposed framework, we use SCAD regression with polynomial basis functions with degree 2 for the simulated data and machine learning approaches including Xgboost (Chen and Guestrin, 2016) and random forest (Breiman, 2001) for the application.

We evaluate four categories of methods:

- Outcome adjusted methods as introduced in Section 5.1 based on either the RCT, OS or integrated data, denoted as "*rct_oa*", "*os_oa*" or "*int_oa*", respectively.

- $R$-learner methods based on either the RCT, OS or integrated data, denoted as "*rct_rlearner*", "*os_rlearner*", "*naive_rlearner*" (standard $R$-learner assuming no hidden confounding) or "*int_rlearner*" (our method), respectively.

- Learning two regression functions with causal forest for the treated and control and taking their difference using either the RCT or OS, denoted as "*rct_rf*" or "*os_rf*", respectively.

- The method proposed in Kallus et al. (2018) where the confounding function is estimated by linear regression or lasso regression with polynomial basis functions, denoted as "*KPS_linear*" or "*KPS_lasso*", respectively.

The performances are measured by the square root of the mean square errors (RMSE) between the estimated condition treatment effect and the ground truth over the test data, i.e. $\text{RMSE}(\hat{\tau}) = \sqrt{T^{-1} \sum_{t=1}^{T} \{\hat{\tau}(X_{\text{test}}^{(t)}) - \tau(X_{\text{test}}^{(t)})\}^2}$, where $T$ is the sample size of the test data.

## 6.1. Simulation study

We modify two simulation settings considered in the literature. For the first set of simulation studies, we consider the simulation setting with scalar covariates as in Kallus et al. (2018), where the RCT and the OS have only partial overlap and the true confounding function is linear. For the second set of simulation studies, we follow the simulation setting with multiple covariates in Yang et al. (2020c) where the true confounding function is non-linear and the strength of confounding bias can be adjusted.

### 6.1.1. SIMULATION STUDY I

We generate the RCT data with sample size $n$ as follows:

$$X_{\text{rct}} \sim \text{Uniform}[-1, 1], U_{\text{rct}} \sim \mathcal{N}(0, 1), A_{\text{rct}} \sim \text{Bernoulli}(0.5).$$

Next we generate the OS data with sample size $m = 2000$ as follows: $A_{\text{os}} \sim \text{Bernoulli}(0.5)$, and the observed covariate $X_{\text{os}}$ and the unobserved covariate $U_{\text{os}}$ are sampled from a bivariate Normal distribution with sample size $m$

$$\begin{pmatrix} X_{\text{os}} \\ U_{\text{os}} \end{pmatrix} \mid A_{\text{os}} \sim \mathcal{N} \left\{ \begin{pmatrix} 0 \\ 0 \end{pmatrix}, \begin{pmatrix} 1 & A_{\text{os}} - 0.5 \\ A_{\text{os}} - 0.5 & 1 \end{pmatrix} \right\}.$$

For both datasets, we generate the outcomes $Y = 1 + X + 0.5X^2 + U + A\tau(X) + 0.5\epsilon$, where $\tau(X) = 1 + 2X + 0.75X^2$ and $\epsilon \sim \mathcal{N}(0, 1)$. Then by the definition of the confounding function, we can get $c(X) = \mathbb{E}(U \mid X, A = 1, S = 0) - \mathbb{E}(U \mid X, A = 0, S = 0) = X$. The test data $X_{\text{test}}$ is sampled from $Uniform[-1, 1]$ with size $T = 10^5$. The experiment results of $\text{RMSE}(\hat{\tau})$ for different methods with the increasing sample sizes of the RCT $n$ are presented in Table 1. The results of the methods with only using the OS are unchanged across the rows due to $m$ unchanged and thus are left blank, and the RMSEs of the other methods are decreasing with $n$ increasing, which empirically verifies their consistency. Among all the methods, our proposed method has the lowest or at least competitively lowest RMSEs. "*KPS_linear*" gains the improvement compared with "*rct_rf*" and "*os_rf*" due to the correctly specified confounding function, but since its consistency at a relatively slow rate based on random forest, it is hard to defeat the outcome-adjusted method and the integrative $R$-learner. Besides, "*naive_rlearner*" performs badly due to ignoring the unmeasured confounding bias of the OS. Finally, "*int_oa*" has pretty decent performance in such a low dimensional setting.

Table 1: The averaged RMSE($\hat{\tau}$) over 100 experiment replicates (standard error in parentheses).

| n | rct_oa | os_oa | int_oa | rct_rlearner | os_rlearner | int_rlearner | rct_rf | os_rf | KPS_lasso | KPS_linear | naive_rlearner |
|---|---|---|---|---|---|---|---|---|---|---|---|
| 200 | 0.28 | | 0.29 | 0.26 | | **0.23** | 0.38 | | 0.43 | 0.32 | 0.56 |
| | (0.01) | | (0.02) | (0.01) | | (0.01) | (0.01) | | (0.01) | (0.01) | (0.00) |
| 500 | 0.16 | 0.57 | 0.13 | 0.15 | 0.58 | **0.11** | 0.25 | 0.62 | 0.29 | 0.25 | 0.53 |
| | (0.01) | (0.00) | (0.01) | (0.01) | (0.00) | (0.01) | (0.01) | (0.01) | (0.01) | (0.01) | (0.00) |
| 1000 | 0.11 | | **0.08** | 0.11 | | 0.09 | 0.23 | | 0.25 | 0.22 | 0.50 |
| | (0.00) | | (0.00) | (0.00) | | (0.01) | (0.00) | | (0.01) | (0.00) | (0.00) |

### 6.1.2. SIMULATION STUDY II

We first generate a population of size $T = 10^5$ with independent $p$-dimensional covariates $X_j \sim \mathcal{N}(0,1)$, $j = 1, \ldots, p$. Let the treatment effect modifier be $W = (X_1, X_2, \ldots, X_{p-1})$, where $p = 10$. Then we generate the potential outcomes $Y(a)$ by $Y(a) \mid X = X_1 + X_p + a\tau(W) + \epsilon(a)$, where $\epsilon(a) \sim \mathcal{N}(0,1)$, for $a = 0, 1$. Now we generate the RCT: the RCT selection indicator is generated by $S \mid X \sim \text{Bernoulli}\{\pi_S(X)\}$, where $\text{logit}\{\pi_S(X)\} = -7 - X_1 + X_2$, which results in around 200 RCT samples ($S = 1$). Then an OS sample is simply randomly selected from the population with size $m = 500$. The treatments are assigned as follows: $A \mid X, S = 1 \sim \text{Bernoulli}\{e(X, S = 1)\}$, where $\text{logit}\{e(X, S = 1)\} = -1 - X_1 + X_2$, and $A \mid X, S = 0 \sim \text{Bernoulli}\{e(X, S = 0)\}$, where $\text{logit}\{e(X, S = 0)\} = -1 - X_1 + X_2 + bX_p$. We let $X_p$ is unobserved in both the RCT and the OS, thus $b$ indicates the strength of unmeasured confounding: the larger absolute value of $b$ induces the larger confounding bias. We vary $b \in \{0, 2.5\}$, and the confounding function equals to zero when $b = 0$. The observed outcomes $Y = AY(1) + (1 - A)Y(0)$. We also consider different dimensions of the treatment modifiers by letting $\tau(W) = 1 + \sum_{j=1}^{J} X_j, J \in \{1, 9\}$. The test data $X_{\text{test}}$ is all the first $p - 1$ dimensional covariates in the initially generated population of size $T = 10^5$. The results of RMSEs for the different scenarios are presented in Table 2. For the same $J$, the results of the methods with only using the RCT are unchanged and thus are left blank. As shown in Table 2, when $b$ increasing, the RMSEs of methods with the OS alone will increase. The proposed integrative $R$-learner performs the best among all the methods. *"KPS_linear"* suffers high loss due to its incorrectly specified model for the non-linear $c(X)$ and *"int_oa"* also perform badly due to the instability induced from the inverse of the propensity scores estimated by the high dimensional covariates.

Table 2: The averaged RMSE($\hat{\tau}$) over 100 experiment replicates (standard error in parentheses).

| J | b | rct_oa | os_oa | int_oa | rct_rlearner | os_rlearner | int_rlearner | rct_rf | os_rf | KPS_lasso | KPS_linear | naive_rlearner |
|---|---|---|---|---|---|---|---|---|---|---|---|---|
| 1 | 0 | | 0.55 | 0.86 | | 0.49 | **0.28** | | 0.75 | 0.99 | 1.45 | 0.41 |
| | | 1.12 | (0.02) | (0.06) | 0.88 | (0.02) | (0.03) | 1.19 | (0.01) | (0.03) | (0.12) | (0.01) |
| | 2.5 | (0.14) | 1.40 | 1.35 | (0.04) | 1.33 | **0.48** | (0.02) | 1.31 | 0.83 | 1.44 | 1.01 |
| | | | (0.01) | (0.05) | | 1.33(0.01) | (0.04) | | (0.01) | (0.02) | (0.11) | (0.01) |
| 9 | 0 | | 1.08 | 1.45 | | 0.97 | **0.55** | | 2.33 | 2.22 | 1.75 | 0.77 |
| | | 1.87 | (0.03) | (0.08) | 1.64 | (0.02) | (0.03) | 2.54 | (0.01) | (0.03) | (0.13) | (0.01) |
| | 2.5 | (0.13) | 1.55 | 1.95 | (0.03) | 1.54 | **1.12** | (0.01) | 2.63 | 2.18 | 1.70 | 1.31 |
| | | | (0.02) | (0.08) | | (0.01) | (0.05) | | (0.01) | (0.03) | (0.12) | (0.01) |

## 6.2. Real data application

The STAR (Tennessee Student/Teacher Achievement Ratio) experiment is a randomized controlled experiment aiming to study the effect of class size on students' standardized test scores. The treatments are two types of class ( $A = 1$ for small classes containing 13-17 pupils and $A = 0$ for regular

classes containing 22-25 pupils). The outcome $Y$ is the sum of the math, reading, and listening standardized test scores. The vector of covariates $X$ includes gender, race, birth month, birthday, birth year, an indicator of giving free lunch or not, and teacher id. Among 4218 students, 2413 were randomly assigned to regular-size classes ($A = 0$) and 1805 to small classes ($A = 1$).

Following Kallus et al. (2018), we create synthetic RCT, OS, and test data from the original data. Split all samples over a binary variable $U$: rural or inner-city ($U = 1$; 2811 students) vs. urban or suburban ($U = 0$; 1407 students), which is known to strongly affect outcome (Krueger, 1999). The RCT is formed by randomly sampling half students with $U = 1$. The OS consists of two parts. (a) From samples with $U = 1$, take samples with $A = 0$ but not in the RCT and samples with $A = 1$ whose outcomes are in the lower half of outcomes among samples with $A = 1$ & $U = 1$; (b) From samples with $U = 0$, take all samples with $A = 0$ and samples with $A = 1$ whose outcomes are in the lower half of outcomes among samples with $A = 1$ & $U = 0$. This procedure results in the RCT and the OS not fully overlapping since the RCT only has rural or inner-city students, and it also biases the treatment effect estimates in the OS downward since only the lower half of scores among samples with $A = 1$ are selected into the OS. Let $U$ be the unmeasured confounder, the outcome and treatment in the OS are confounded significantly. Since the ground truth, $\tau_i$ is inaccessible in practice, it is replaced with an unbiased estimate from all samples, which is defined as $A_i\{Y_i - \mathbb{E}(Y_i)\}/P(A_i = 1) - (1 - A_i)\{Y_i - \mathbb{E}(Y_i)\}/\{1 - P(A_i = 1)\}$, $i \in$ all samples, where $\mathbb{E}(Y_i)$ is estimated by the average of the outcomes and $P(A_i = 1)$ is estimated by the proportion of treated samples among all samples. The test data is the held-out sample of all samples excluding the RCT.

To demonstrate the generality of the proposed framework, we implement the integrative $R$-learner with start-of-art machine learning methods (Xgboost and random forest (Breiman, 2001)) for approximating the nuisance functions. An iterative learning procedure is provided in Appendix D to realize different penalties on $\tau(\cdot)$ and $c(\cdot)$. We initialize the values of the confounding function as the ones estimated from the method in Kallus et al. (2018), iterate the algorithm for 20 times, and calculate the RMSE. Figure 1 displays the results over iterations. As the number of iterations increases, the RMSE of our integrative $R$-learner reaches a steady value. The estimators with only the RCT or the OS give the RMSE around 80 (not shown in Figure 1), the method in Kallus et al. (2018) has a significant improvement (RMSE = 71.01), and the proposed method has the best performance (RMSE $\approx$ 70).

## 7. Discussion

We propose the integrative $R$-learner which can identify both the HTE and confounding function, and improve the efficiency of the RCT estimator of the HTE. It is a general framework that has a broad class of choices for employing flexible models in either nuisance function approximation or loss-minimization procedures. The proposed method is motivated in the settings where the sample size of the RCT ($n$) is small but the sample size of the OS ($m$) is much larger. To address the practical concern, we can allow $n/m$ to go to 0 with $n$ and $m$ going to infinity in our asymptotic regime. However, the non-asymptotic risk bounds for a fixed $n$ and a growing $m$ are also of great interest in practice, whose derivations will be presented elsewhere. Moreover, in practice, both RCT and OS data are likely to present missing values, requiring proper assumptions and statistical analysis methods for handling missing data (Yang et al., 2019). Multiple imputation is a popular and intuitive approach that fills missing values by plausible values multiple times and applies complete-sample analysis methods to each imputed data set. Following Guan and Yang (2019), one can

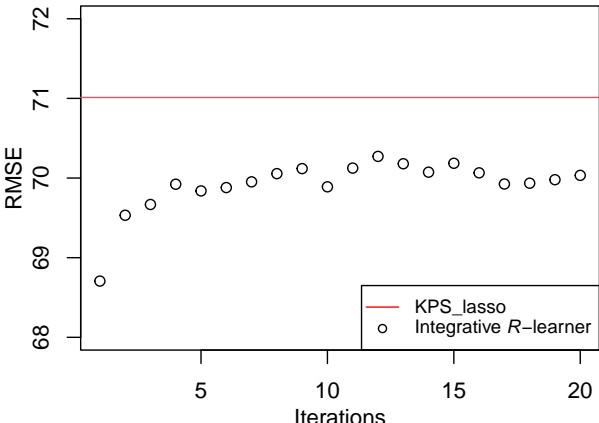

Figure 1: The plot of RMSE for the integrative $R$-learner at each iteration vs. RMSE of the estimator proposed in Kallus et al. (2018)

incorporate the idea of multiple imputation in the proposed framework to handle missing values in the RCT and OS.

There are interesting directions for future work. Besides the continuous outcomes discussed in the paper, the proposed framework can be readily extended to other types of outcomes, e.g., a binary outcome (Vansteelandt and Joffe, 2014) or a survival outcome (Yang et al., 2020a; Yang, 2021). Another interesting direction is to extend the integrative $R$-learner to multiple treatments which commonly arise in reality. It can be constructed similarly based on the multivariate version of Robinson's transformation and the corresponding $R$-learner (Nie and Wager, 2021). Besides, it would also be interesting to extend the integrative $R$-learner to off-policy evaluation under covariate shifts (Uehara et al., 2020) or generalizable individualized decision rules learning (Zhao et al., 2019; Wu and Yang, 2021; Chu et al., 2022) to improve the efficiency of the estimated policy values or the estimated optimal decision rules.

## Acknowledgments

Yang is partially supported by the NSF DMS 1811245, NIH 1R01AG066883 and 1R01ES031651.

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

# Supplementary Material for "Integrative $R$-learner of heterogeneous treatment effects combining experimental and observational studies" by Wu and Yang

## Outline of the appendices

- Appendix A: a pseudocode of the penalty scale searching for the integrative $R$-learner.

- Appendix B: a proof of Theorem 1.

- Appendix C: a proof of Proposition 2.

- Appendix D: a pseudocode of iterative learning algorithm used in the real data experiment.

## Appendix A. The algorithm of the penalty scale searching for the integrative $R$-learner introduced in Section 3.3

---
**Algorithm 1** Penalty scale searching for the integrative $R$-learner

---
**Data:** The RCT and the OS data $\{(X_i, A_i, Y_i, S_i)\}_{i \in \mathcal{I}_n \cup \mathcal{I}_m}$; the estimated nuisance functions $\hat{\mu}^{-k(i)}(X_i, S_i)$ and $\hat{e}^{-k(i)}(X_i, S_i), i \in \mathcal{I}_n \cup \mathcal{I}_m$

Initialize $cmv = \infty$

**Result:** $\beta_\tau$ and the HTE $\tau(\cdot) = p_\tau(\cdot)^\mathsf{T} \beta_\tau$

**for** $sc = 0, 0.5, 1, 1.5$ **do**

    Fit $Y_i - \hat{\mu}^{-k(i)}(X_i, S_i)$ on $\{p_\tau(X_i), (1 - S_i)p_c(X_i)\}\{A_i - \hat{e}^{-k(i)}(X_i, S_i)\}$ with penalized regression where the corresponding shrinkage parameters are $(sc\lambda_c, \lambda_c)$ and $\lambda_c$ is chosen by cross-validation;

    Record the above mean cross-validated error $cvm_{sc}$ and the fitted coefficients $\beta_{sc,\tau}$;

    **if** $cvm_{sc} < cvm$ **then**

        $\beta_\tau \leftarrow \beta_{sc,\tau}$

        $cvm \leftarrow cvm_{sc}$

    **end**

**end**

---

## Appendix B. Proof of Theorem 1

To prove Theorem 1, we show a useful lemma first.

**Lemma 3** *Under Assumptions 3-4, $\widehat{L}_N(\beta) = L_N(\beta) + O_P(a_N^2)$.*

**Proof** For the simplicity of the notations, we denote

$$A_{\mu,i} = \mu(X_i, S_i) - \hat{\mu}^{-k(i)}(X_i, S_i), \ B_{\mu,i} = Y_i - \mu(X_i, S_i),$$
$$A_{e,i} = e(X_i, S_i) - \hat{e}^{-k(i)}(X_i, S_i), \ B_{e,i} = A_i - e(X_i, S_i).$$

By algebra, we have

$$\widehat{L}_N(\beta) = \mathbb{P}_N\{B_{\mu,i} + A_{\mu,i} - g(X_i, S_i; \beta)B_{e,i} - g(X_i, S_i; \beta)A_{e,i}\}^2$$

$$
\begin{aligned}
=&\mathbb{P}_N\{B_{\mu,i} - g(X_i, S_i; \beta)B_{e,i}\}^2 + \mathbb{P}_N\{A_{\mu,i} - g(X_i, S_i; \beta)A_{e,i}\}^2 + \\
&2\mathbb{P}_N\{B_{\mu,i} - g(X_i, S_i; \beta)B_{e,i}\}\mathbb{P}_N\{A_{\mu,i} - g(X_i, S_i; \beta)A_{e,i}\} \\
=&L_N(\beta) + \mathbb{P}_N A_{\mu,i}^2 + \mathbb{P}_N A_{e,i}^2 g^2(X_i, S_i) - 2\mathbb{P}_N A_{\mu,i}A_{e,i}g(X_i, S_i; \beta) + \\
&2\mathbb{P}_N B_{\mu,i}A_{\mu,i} - 2\mathbb{P}_N B_{\mu,i}A_{e,i}g(X_i, S_i; \beta) - 2\mathbb{P}_N B_{e,i}A_{\mu,i}g(X_i, S_i; \beta) + 2\mathbb{P}_N B_{e,i}A_{e,i}g^2(X_i, S_i).
\end{aligned}
$$

By Markov's inequality and Assumption 4, we have the second term above $\mathbb{P}_N A_{\mu,i}^2$ is $O_P(a_N^2)$. Plus Assumption 3, we have the third term $\mathbb{P}_N A_{e,i}^2$ is $O_P(a_N^2)$. As for the fourth term,

$$
\mathbb{P}_N A_{\mu,i}A_{e,i}g(X_i, S_i; \beta) \leq C_1 \mathbb{P}_N A_{\mu,i}A_{e,i} \leq C_1\sqrt{\mathbb{P}_N A_{\mu,i}^2 \mathbb{P}_N A_{e,i}^2} = O_P(a_N^2),
$$

for some positive constant $C_1$, and the last inequality is by Cauchy-Schwarz inequality.

Next, to deal with the last four terms, we tackle $\mathbb{P}_N B_{\mu,i}A_{\mu,i}$ first. Let

$$
B_{\mu\mu}^k = \frac{\sum_{i:k(i)=k} B_{\mu,i}A_{\mu,i}}{|\{i : k(i) = k\}|},
$$

and note that $|\mathbb{P}_N B_{\mu,i}A_{e,i}| \leq \sum_{k=1}^K |B_{\mu\mu}^k|$, where $K$ is finite, thus it is suffice to show $B_{\mu\mu}^k = O_P(a_N^2)$. Let $\mathcal{I}^{-k} = \{X_i, A_i, Y_i, S_i : k(i) \neq k\}$. Then we have

$$
\begin{aligned}
\mathbb{E}(B_{\mu\mu}^k) = \mathbb{E}(B_{\mu,i}A_{e,i}) &= \mathbb{E}\{\mathbb{E}(B_{\mu,i}A_{e,i} \mid \mathcal{I}^{-k}, X_i, S_i)\} \\
&= \mathbb{E}\{A_{e,i}\mathbb{E}(B_{\mu,i} \mid \mathcal{I}^{-k}, X_i, S_i)\} = 0.
\end{aligned}
$$

Then we have its variance

$$
\begin{aligned}
\mathrm{var}(B_{\mu\mu}^k) = \mathbb{E}\{(B_{\mu\mu}^k)^2\} &= \frac{\mathbb{E}\{\sum_{i:k(i)=k} B_{\mu,i}^2 A_{\mu,i}^2 + \sum_{i\neq j:k(i)=k,k(j)=k} B_{\mu,i}B_{\mu,j}A_{\mu,i}A_{\mu,j}\}}{|\{i : k(i) = k\}|^2} \\
&= \frac{\mathbb{E}(B_{\mu,i}^2 A_{\mu,i}^2)}{|\{i : k(i) = k\}|} + \frac{\sum_{i\neq j:k(i)=k,k(j)=k} \mathbb{E}(B_{\mu,i}B_{\mu,j}A_{\mu,i}A_{\mu,j})}{|\{i : k(i) = k\}|^2}.
\end{aligned}
$$

By Assumption 3-4 we have $\mathbb{E}(B_{\mu,i}^2 A_{e,i}^2) = \mathbb{E}\{\mathbb{E}(B_{\mu,i}^2 A_{e,i}^2 \mid \mathcal{I}^{-k}, X_i, S_i)\} = \mathbb{E}\{A_{e,i}^2 \mathbb{E}(B_{\mu,i}^2 \mid \mathcal{I}^{-k}, X_i, S_i)\} \leq C_2 \mathbb{E}A_{e,i}^2 = O(a_N^2)$, for some positive constant $C_2$. As for the interaction terms, we have $\mathbb{E}(B_{\mu,i}B_{\mu,j}A_{\mu,i}A_{\mu,j}) = \mathbb{E}\{A_{\mu,i}A_{\mu,j}\mathbb{E}(B_{\mu,i}B_{\mu,j} \mid \mathcal{I}^{-k}, X_i, S_i)\} = \mathbb{E}\{A_{\mu,i}A_{\mu,j}\mathbb{E}(B_{\mu,j})\mathbb{E}(B_{\mu,i} \mid \mathcal{I}^{-k}, X_i, S_i)\} = 0$. The second last equality is implied by $B_{\mu,i}$ and $B_{\mu,j}$ are independent for $i \neq j$, and the last equality comes from the definition of $B_{\mu,i}$. Therefore, we have $\mathrm{var}(B_{\mu\mu}^k) = (K/N)O(a_N^2) = O(a_N^2/N)$, which is negligible with a faster diminishing rate than $O(a_N^2)$. Then, by Chebyshev' inequality, we have $B_{\mu\mu}^k = O_P(a_N^2/N)$, i.e., $\mathbb{P}_N B_{\mu,i}A_{\mu,i} = O_P(a_N^2/N)$. Similarly, under the Assumption 3 that $g(X_i, S_i; \beta)$ is uniformly bounded, we can get the same results for the left three terms equal to $O_P(a_N^2/N)$. Therefore, $\widehat{L}_N(\beta) - L_N(\beta)$ is dominated by the $O_P(a_N^2)$-term $\mathbb{P}_N A_{\mu,i}^2 + \mathbb{P}_N A_{e,i}^2 g^2(X_i, S_i) - 2\mathbb{P}_N A_{\mu,i}A_{e,i}g(X_i, S_i; \beta)$, which finally leads to the conclusion in the lemma $\widehat{L}_N(\beta) - L_N(\beta) = O_P(a_N^2)$. ∎

Next, we are showing the proof of Theorem 1 with the help of Lemma 3.

**Proof of Theorem 1:**

First, we let $\mathbb{G}_N = \sqrt{N}(\mathbb{P}_N - \mathbb{P})$, $p^\tau = \{A - e(X,A)\}p_\tau(X)$, and $p^c = \{A - e(X,A)\}(1-S)p_c(X)$. Recall $p_i^\intercal = \{(p^\tau)^\intercal, (p^c)^\intercal\}$, and then

$$\Gamma = \mathbb{P}(p_i p_i^\intercal) = \begin{Bmatrix} p^\tau(p^\tau)^\intercal & p^\tau(p^c)^\intercal \\ p^c(p^\tau)^\intercal & p^c(p^c)^\intercal \end{Bmatrix} \overset{\text{denoted as}}{=} \begin{pmatrix} \Gamma_{\tau\tau} & \Gamma_{\tau c} \\ \Gamma_{c\tau} & \Gamma_{cc} \end{pmatrix}.$$

By the definition of $\hat{\beta}$ and Lemma 3, we have

$$\hat{\beta} = \underset{b}{\operatorname{argmin}} \, \mathbb{P}_N \left\{Y_i - \mu(X_i, S_i) - p_i^\intercal b\right\}^2 + O_P(a_N^2).$$

By the Pointwise Linearizaiton of the series method (see Lemma 4.1 in Belloni et al., 2015), under Assumptions 5-7 and the fact $\mathbb{E}(\epsilon \mid X, A, S) = 0$ which is shown under Assumptions 1-2 in Section 3, we have for any $\alpha := (\alpha_\tau^\intercal, \alpha_c^\intercal)^\intercal \in \mathbb{R}^{2D}$,

$$\sqrt{N}\alpha^\intercal(\hat{\beta} - \beta) = \alpha^\intercal \begin{pmatrix} \Gamma_{\tau\tau} & \Gamma_{\tau c} \\ \Gamma_{c\tau} & \Gamma_{cc} \end{pmatrix}^{-1} \mathbb{G}_N \begin{pmatrix} p^\tau\epsilon \\ p^c\epsilon \end{pmatrix} + o_P(1) + O_P\left(\sqrt{N}a_N^2\right). \qquad \text{(S1)}$$

Let

$$\Sigma = \begin{pmatrix} \Sigma_{\tau\tau} & \Sigma_{\tau c} \\ \Sigma_{c\tau} & \Sigma_{cc} \end{pmatrix} := \begin{pmatrix} \Gamma_{\tau\tau} & \Gamma_{\tau c} \\ \Gamma_{c\tau} & \Gamma_{cc} \end{pmatrix}^{-1}.$$

Since under Assumption 4, $a_N = O(N^{-r}), r > 1/4$, thus $O_P\left(\sqrt{N}a_N^2\right)$ is negligible compared to $o_p(1)$. Therefore, we have

$$\sqrt{N}\alpha_\tau^\intercal(\hat{\beta}_\tau - \beta_\tau) = \alpha_\tau^\intercal \mathbb{G}_N \left(\Sigma_{\tau\tau}p^\tau\epsilon + \Sigma_{\tau c}p^c\epsilon\right) + o_P(1). \qquad \text{(S2)}$$

Under Assumptions 5-7 and the fact $\mathbb{E}(\epsilon \mid X, A, S) = 0$, by the Pointwise Normality of the series method (see Theorem 4.2 in Belloni et al., 2015), we have

$$\sqrt{N}\frac{\alpha_\tau^\intercal(\hat{\beta}_\tau - \beta_\tau)}{\|\alpha_\tau^\intercal\Omega^{1/2}\|} \overset{d}{\to} \mathcal{N}(0,1) + o_P(1),$$

where $\Omega = \mathbb{E}\left\{(\Sigma_{\tau\tau}p^\tau\epsilon + \Sigma_{\tau c}p^c\epsilon)(\Sigma_{\tau\tau}p^\tau\epsilon + \Sigma_{\tau c}p^c\epsilon)^\intercal\right\}$. Then, we take $\alpha_\tau = p^\tau$, for any $x \in \mathcal{X}$,

$$\sqrt{N}\frac{(p^\tau)^\intercal(\hat{\beta}_\tau - \beta_\tau)}{\|(p^\tau)^\intercal\Omega^{1/2}\|} \overset{d}{\to} \mathcal{N}(0,1) + o_P(1).$$

Under Assumption 6(d), the approximation error is negligible relative to the estimation error, then

$$\sqrt{N}\frac{\hat{\tau}(x) - \tau(x)}{\|(p^\tau)^\intercal\Omega^{1/2}\|} \overset{d}{\to} \mathcal{N}(0,1) + o_P(1),$$

which immediately arrives at the first part conclusion in Theorem 1, $\hat{\tau}(x) - \tau(x) = O(N^{-1/2})$, for any $x \in \mathcal{X}$. Besides, it also gives the asymptotic variance of $\hat{\tau}(x)$,

$$\mathbb{V}\{\hat{\tau}(x)\} = N^{-1}(p^\tau)^\intercal\Omega p^\tau. \qquad \text{(S3)}$$

Expanding $\Omega$, under Assumption 7, we have

$$\Omega = \Sigma_{\tau\tau}\mathbb{E}\left\{p^\tau(p^\tau)^\intercal\epsilon^2\right\}\Sigma_{\tau\tau} + \Sigma_{\tau c}\mathbb{E}\left\{p^\tau(p^c)^\intercal\epsilon^2\right\}\Sigma_{\tau c} +$$

$$\Sigma_{\tau c}\mathbb{E}\left\{p^c(p^c)^{\mathsf{T}}\epsilon^2\right\}\Sigma_{\tau c} + \Sigma_{\tau\tau}\mathbb{E}\left\{p^{\tau}(p^c)^{\mathsf{T}}\epsilon^2\right\}\Sigma_{\tau c}$$

$$= \left(\Sigma_{\tau\tau}\Gamma_{\tau\tau} + \Sigma_{\tau c}\Gamma_{\tau c}^{\mathsf{T}}\right)\Sigma_{\tau\tau}\sigma^2 + \left(\Sigma_{\tau c}\Gamma_{cc} + \Sigma_{\tau\tau}\Gamma_{\tau c}\right)\Sigma_{\tau c}^{\mathsf{T}}\sigma^2$$

$$= \Sigma_{\tau\tau}\sigma^2 + \mathbf{0}_{D\times D}$$

$$= \left(\Gamma_{\tau\tau} - \Gamma_{\tau c}\Gamma_{cc}^{-1}\Gamma_{\tau c}^{\mathsf{T}}\right)^{-1}\sigma^2$$

$$= \left[\mathbb{E}\{Sp^{\tau}(p^{\tau})^{\mathsf{T}}\} + \mathbb{E}\{(1-S)p^{\tau}(p^{\tau})^{\mathsf{T}}\} - \Gamma_{\tau c}\Gamma_{cc}^{-1}\Gamma_{\tau c}^{\mathsf{T}}\right]^{-1}\sigma^2. \tag{S4}$$

Next, we aim to obtain the asymptotic variance of the HTE estimator $\hat{\tau}_{\text{rct}} = (p^{\tau})^{\mathsf{T}}\hat{\beta}_{\text{rct}}$ with only the RCT, where $\hat{\beta}_{\text{rct}} = \text{argmin}_{b\in\mathbb{R}^D}\,\mathbb{P}_N S_i\left[Y_i - \hat{\mu}^{-k(i)}(X_i, S_i) - \{A_i - \hat{e}^{-k(i)}(X_i, S_i)\}p_{\tau}^{\mathsf{T}}(X_i)b\right]^2$. Similarly, we can replace the estimated nuisance functions with the true ones based on Lemma 3, and following the same strategy as the integrative $R$-learner above to obtain the asymptotic variance, $\mathbb{V}(\hat{\tau}_{\text{rct}}) = N^{-1}(p^{\tau})^{\mathsf{T}}\Omega_{\text{rct}}p^{\tau}$, where

$$\Omega_{\text{rct}} = \left[\mathbb{E}\left\{Sp^{\tau}(p^{\tau})^{\mathsf{T}}\right\}\right]^{-1}\sigma^2. \tag{S5}$$

By Hölder's inequality, $\mathbb{E}\{(1-S)p^{\tau}(p^{\tau})^{\mathsf{T}}\} - \Gamma_{\tau c}\Gamma_{cc}^{-1}\Gamma_{\tau c}^{\mathsf{T}}$ is non-negative definitive; i.e., for any $v\in\mathbb{R}^D$,

$$v^{\mathsf{T}}(\Omega^{-1} - \Omega_{\text{rct}}^{-1})v \geq 0, \tag{S6}$$

where the inequality becomes an equality if and only if $p_{\tau}(X) = Mp_c(X)$ for some constant matrix $M$. From (S3), we have

$$p^{\tau}\mathbb{V}\{\hat{\tau}(x)\}(p^{\tau})^{\mathsf{T}} = N^{-1}p^{\tau}(p^{\tau})^{\mathsf{T}}\Omega p^{\tau}(p^{\tau})^{\mathsf{T}}$$

$$\implies \mathbf{I}_D\mathbb{V}\{\hat{\tau}(x)\} = N^{-1}p^{\tau}(p^{\tau})^{\mathsf{T}}\Omega$$

$$\implies \mathbf{I}_D\mathbb{V}^{-1}\{\hat{\tau}(x)\} = N\Omega^{-1}\left\{p^{\tau}(p^{\tau})^{\mathsf{T}}\right\}^{-1}$$

$$\implies (p^{\tau})^{\mathsf{T}}\left\{p^{\tau}(p^{\tau})^{\mathsf{T}}\right\}^{-1}\mathbb{V}^{-1}\{\hat{\tau}(x)\}p^{\tau} = N(p^{\tau})^{\mathsf{T}}\left\{p^{\tau}(p^{\tau})^{\mathsf{T}}\right\}^{-1}\Omega^{-1}\left\{p^{\tau}(p^{\tau})^{\mathsf{T}}\right\}^{-1}p^{\tau}$$

$$\overset{\text{by (S6)}}{\implies}(p^{\tau})^{\mathsf{T}}\left\{p^{\tau}(p^{\tau})^{\mathsf{T}}\right\}^{-1}\left[\mathbb{V}^{-1}\{\hat{\tau}(x)\} - \mathbb{V}^{-1}\{\hat{\tau}_{\text{rct}}(x)\}\right]p^{\tau} \geq 0$$

(Multiply a positive number $(p^{\tau})^{\mathsf{T}}p^{\tau}$ on the both sides and get the below formula)

$$\implies (p^{\tau})^{\mathsf{T}}p^{\tau}(p^{\tau})^{\mathsf{T}}\left\{p^{\tau}(p^{\tau})^{\mathsf{T}}\right\}^{-1}p^{\tau}\left[\mathbb{V}^{-1}\{\hat{\tau}(x)\} - \mathbb{V}^{-1}\{\hat{\tau}_{\text{rct}}(x)\}\right] \geq 0$$

$$\implies (p^{\tau})^{\mathsf{T}}p^{\tau}\left[\mathbb{V}^{-1}\{\hat{\tau}(x)\} - \mathbb{V}^{-1}\{\hat{\tau}_{\text{rct}}(x)\}\right] \geq 0$$

$$\implies \mathbb{V}^{-1}\{\hat{\tau}(x)\} - \mathbb{V}^{-1}\{\hat{\tau}_{\text{rct}}(x)\} \geq 0$$

$$\implies \mathbb{V}\{\hat{\tau}(x)\} \leq \mathbb{V}\{\hat{\tau}_{\text{rct}}(x)\},$$

with the equality holding when $p_{\tau}(X) = Mp_c(X)$ for some constant matrix $M$. ∎

## Appendix C. Proof of Proposition 2

**Proof** The proof of (10) is mainly based on the inverse probability weights (IPW) component, thus we tackle it first.

a) IPW-adjusted outcomes: We have

$$\mathbb{E}\left\{\frac{AY}{e(X,S)}\mid X, S = 0\right\}$$

$$= \frac{1}{e(X,0)} [\mathbb{E}\{AY \mid X, A = 1, S = 0\}e(X,0) + \mathbb{E}\{AY \mid X, A = 0, S = 0\}\{1 - e(X,0)\}]$$
$$= \mathbb{E}\{Y \mid X, A = 1, S = 0\}.$$

Similarly, we have

$$\mathbb{E}\left\{ \frac{(1-A)Y}{1 - e(X,S)} \mid X, S = 0 \right\} = \mathbb{E}\{Y \mid X, A = 0, S = 0\}.$$

b) Augmented IPW-adjusted outcomes: Then we have

$$\mathbb{E}\left[ \frac{A\{Y - Q_S(X,1)\}}{e(X,S)} + Q_S(X,1) \mid X, S = 0 \right]$$
$$= \mathbb{E}\left\{ \frac{AY}{e(X,S)} \mid X, S = 0 \right\} + Q_0(X,1)\mathbb{E}\left[ \left\{ 1 - \frac{A}{e(X,S)} \right\} \mid X, S = 0 \right]$$
$$= \mathbb{E}(Y \mid X, A = 1, S = 0).$$

Similarly, we have

$$\mathbb{E}\left[ \frac{(1-A)\{Y - Q_S(X,0)\}}{1 - e(X,S)} + Q_S(X,0) \mid X, S = 0 \right] = \mathbb{E}(Y \mid X, A = 0, S = 0).$$

Finally, by taking the difference of the above two formulas and based on the definition of $c(X)$ in (1), we arrive at the conclusion $\mathbb{E}(\widetilde{Y} \mid X, S = 0) = \tau(X) + c(X)$ ∎

## Appendix D. Iterative learning for the integrative $R$-learner in the real data experiment

---
**Algorithm 2** Iterative learning for the integrative $R$-learner
---
**Data:** The RCT and the OS data $\{(X_i, A_i, Y_i, S_i)\}_{i \in \mathcal{I}_n \cup \mathcal{I}_m}$; the number of iterations $B$
**Result:** $\tau(\cdot)$
Initialize confounding function $c(X_i)$.
Estimate the nuisance functions with cross-fitting based on Xgboost, denoted as $\hat{\mu}(X_i, S_i)$ and $\hat{e}(X_i.S_i)$, resulting in $\hat{\epsilon}_{Y_i} = Y_i - \hat{\mu}(X_i, S_i)$ and $\hat{\epsilon}_{A_i} = A_i - \hat{e}(X_i, S_i)$.
**for** $b = 1, \ldots, B$ **do**

    Calculate the pseudo outcomes $\widetilde{Y}_i \leftarrow \hat{\epsilon}_{Y_i}/\hat{\epsilon}_{A_i} - (1 - S_i)c(X_i), i \in \mathcal{I}_n \cup \mathcal{I}_m$;
    Obtain $\tau(\cdot)$ by fitting $\widetilde{Y}_i$ on $X_i$ using weighted random forest with weights equal to $\hat{\epsilon}_{A_i}^2, i \in \mathcal{I}_n \cup \mathcal{I}_m$;
    Update the pseudo outcomes $\widetilde{Y}_i \leftarrow \hat{\epsilon}_{Y_i}/\hat{\epsilon}_{A_i} - \tau(X_i), i \in \mathcal{I}_m$;
    Update $c(X_i)$ by fitting $\widetilde{Y}_i$ on $X_i$ using weighted random forest with weights equal to $\hat{\epsilon}_{A_i}^2, i \in \mathcal{I}_m$.

**end**
---

