# OpenReview forum: "Integrative $R$-learner of heterogeneous treatment effects combining experimental and observational studies"
_cclear.cc/CLeaR/2022/Conference — CLeaR 2022 Poster_

### Official Review · Reviewer_YmAo · 2021-11-22

**Confidence:** 3
**Overall Score:** 8

**Main Review:**

Originality:

* The approach leverages the flexibility of the R-learner to use multiple machine learning methods, while also supporting the integration of observational and experimental data to predict treatment effects.

* The approach uses different regularization terms for the HTE and confounding functions.

* The paper includes in the supplement a proof that the integrative R-learner is consistent and that asymptotically it is at least as efficient as the estimator that uses only experimental data.

Significance:

* The problem of leveraging both observational and experimental data to improve the prediction of treatment effects is quite important in many fields. In healthcare, for example, it could support better predictions of treatment outcomes for individual patients, which can help optimize their care.

* The ability to use a wide range of supervised statistical and machine learning methods to address this problem is important, because it allows current and future methods to be immediately applied to help improve the prediction of treatment effects.

* The paper does not discuss the issue of model interpretability. How feasible would it be to develop an automated explanation capability that describes the factors in the model that are driving its predictions? For the real-data experiments described in the paper, what were the factors driving the predictions?

* To what extent are the assumptions made in the simulation studies in alignment with the assumptions made by the Integrative R-Learner method? For the simulation study, did the authors “stress test” their method by making assumptions that are (1) counter to those made by their Integrative R-Learner method, and (2) compatible with the assumptions made by other methods that were evaluated? The paper does not discuss this issue, and thus, it seems that they did not.

* The Discussion section is quite brief. It would be useful to include, for instance, some discussion of the limitations of the described approach and directions for future research. For example, how would the method handle a treatment outcome that is naturally represented using a nominal variable (e.g., a binary variable representing in-hospital death)? How would it handle the common situation in which the covariates in an RCT are a small subset of the available observational covariates in the electronic medical record (EMR) system? The paper doesn’t have to solve these problems of course, but some discussion of such issues and open problems would be helpful.

Technical quality:

* The work looks to be of high quality, but I did not carefully check the proofs.

* In the real experiments, what is the standard error of the predictions shown in Figure 1?

Clarity:

* Overall, the paper is grammatically well written.

* The paper is written in a style that will be easiest for readers with a strong background in mathematical statistics to understand. Several parts of the paper may be difficult for readers with other backgrounds to understand clearly and completely.


**Summary:**

The current paper builds on the R-learner method developed by Nie & Wager. The Nie & Wager method provides a flexible framework for jointly using a variety of statistical and machine learning methods to predict treatment effects from observational data under the assumption of no unmeasured confounding of the treatment and outcome. The current paper develops the Integrative R-learner method which uses both observational (e.g., EMR) and experimental (e.g., RCT) data on a common set of real-valued covariates to predict treatment effects of a binary treatment on a real-valued outcome, when unmeasured confounders are possible. The method uses the experimental data to reduce bias in the causal predictions, and it uses the observational data to reduce variance, when doing so is possible. An evaluation using simulated and real data support that the method predicts treatment effects better than several other state-of-the-art methods.

---

> ### Author Response · Authors · 2021-12-04
> **Responses to Reviewer YmAo**
>
> We thank the reviewer for your careful reading of our manuscript and helpful comments. In what follows, we have the point-to-point responses to the detailed review comments.
>
> 1. *“The paper does not discuss the issue of model interpretability. How feasible would it be to develop an automated explanation capability that describes the factors in the model that are driving its predictions? For the real-data experiments described in the paper, what were the factors driving the predictions?”*
>
>     Thanks for the constructive comment for model interpretability. Yes, it is feasible. For the penalized series approximation such as lasso regression, we can select important variables that drive the heterogeneity of treatment effect, i.e., the variables whose coefficients are not zero. For machine learning such as Xgboost, we can obtain feature importance with built-in functions in the R packages or Python libraries, which are calculated by the out-of-bag estimation error.
>
> 2. *“To what extent are the assumptions made in the simulation studies in alignment with the assumptions made by the Integrative R-Learner method? …, it seems that they did not.”*
>
>     The current simulated data satisfy the required assumptions for the proposed integrative R-learner to be consistent. For the real-data experiment, the causal assumptions, e.g. Assumption 2 (Transportability of the HTE) are not testable empirically and thus rely on subject matter knowledge for their justification. We will design different simulation settings where one or multiple required assumptions are violated to conduct stress tests on the consistency of the proposed estimator.
>
> 3. *“how would the method handle a treatment outcome that is naturally represented using a nominal variable (e.g., a binary variable representing in-hospital death)?”*
>
>     When the outcome is not a continuous variable, e.g., a binary outcome, we can extend the proposed framework to study causal odds ratios instead of average treatment effects (see *Vansteelandt, Stijn, and Marshall Joffe. "Structural nested models and G-estimation: the partially realized promise." Statistical Science 29, no. 4 (2014): 707-731.*)
>
> 4. *“How would it handle the common situation in which the covariates in an RCT are a small subset of the available observational covariates in the electronic medical record (EMR) system?”*
>
>     Thanks for pointing out this important question. Our proposed method can handle the case when the covariates in an RCT capture all the treatment effect modifiers, and the confounding function can incorporate the additional auxiliary variables in the OS. If the treatment effect modifiers are not captured in one of the datasets, we cannot leverage the OS sample to improve the HTE estimation in the RCT sample. That being said, this is an important and interesting question for our future work.
>
> 5. *“In the real experiments, what is the standard error of the predictions shown in Figure 1?”*
>
>     Since the real data is fixed and given, we are not able to repeat the experiments like in the simulation studies, thus standard error is not able to be calculated.

---

> > ### Comment · Reviewer_YmAo · 2021-12-11
> > **Thanks to the authors for their response.**
> >
> > I thank the authors for their helpful response to my comments and questions.

---

### Official Review · Reviewer_8X2B · 2021-11-24

**Confidence:** 3
**Overall Score:** 5

**Main Review:**

### Originality

The approach proposed in the paper seems novel, as far as I can tell. However, I think a few citations to related works might be missing. How does the approach relate to, e.g.

Zhang and Bareinboim (2017) Transfer learning in multi-armed bandits: A causal approach.

Zhang and Bareinboim (2021) Bounding causal effects on continuous outcomes.

### Significance

This paper addresses an important problem, that of merging observational and experimental data in causal inference, which is very relevant to CLeaR. However, it is also very dense mathematically for a quite limited in scope (binary contextual bandits). And most importantly, I remain unconvinced, from the presented experimental results, of the utility of the method. Thus, I am unsure this paper is likely to have a broad impact.

### Technical quality

The technical quality of the paper is rather poor. The presented approach is backed-up by theoretical justifications, however it is hard to judge the validity of those, or how restrictive the required assumptions are. The paper is very dense and mathematically heavy (although I noted the notation was also a bit sloppy at times), and the proposed approach includes many bells and whistles (different models for different parts of the model, cross-fitting, series approximation, regularization with cross-validation) which highly obfuscate the interpretability of the paper. On the experimental side also, I found the experiments rather limited and potentially flawed (see my detailed comments), and thus not convincing enough to support the claims of the paper.

### Clarity

Although the proposed approach does not seem that complicated and can be summarized in two equations (5 and 6), the mathematical developments in the paper are hard to parse and interpret intuitively. Also, I found the experiments not very clear. For example, I could not clearly understand what the compared methods actually correspond to (e.g., the rct_x and os_x variants).

### Detailed comments

p.3 Ass.1: $c_1,c_2 \in (0, 1)$ -> This notation is confusing to me. Do you mean the interval $[0,1]$, or the set $\\{0,1\\}$ ? This should be clarified.

p.3 Ass.1: $\pi_A(X)$ -> this concept has not been introduced yet. What is $\pi_A(X)$ ? Furthermore, if I assume that this represents the policy used in the observational study, $p(A|X,S=0)$, then this statement is trivial. Since $A$ is binary, we necessarily have that $p(A=a|X,S=0) \in [0,1], \forall a \in \\{0,1\\}$.

p.3 Ass.1: I have a hard time understanding the meaning of (c). I suppose you mean $Y=Y(a), \forall a$ ? Then, this turns your assumption (a) into $A \perp Y \mid (X,S=1)$, which means that the treament has no causal effect on the outcome. Am I missing something ?

p.3 Ass.2: $s=0,1$ -> do you mean $\forall s \in \\{0,1\\}$ ?

p.3 eq.2: $A\\{\dots\\}$ -> this notation is again new. What does $A\\{\dots\\}$ mean ?

p.3: combining the RCT and OS can identify the HTE and confounding function -> the first part of this statement is trivial, since the RCT alone can identify the HTE.

p.3: $Y(a) \perp S \mid X,(a=0,1)$ -> this notation again is rather sloppy. Do you mean $Y(a) \perp S \mid X,A=a, \forall a \in {0,1}$ ?

p.3 Ass.2: $\mathbb{E}\\{...\\}=\tau(X)$ -> notation is inconsistent, should be $\mathbb{E}(...)$.

p.6: the possible smooth or sparsity of $c(X)$ leads to the efficiency gain of the integrative R-learner compared to using the RCT alone -> Do the gains of your method only come from regularization ? And what if the RCT $\tau(X)$, estimated alone from RCT data, is also regularized ? Do you still obtain gains with your method ?

p.6: Thus, it is critical to impose different regularization parameters for $\tau(X)$ and $c(X)$ -> I do not see the implication. Why would these regularizations need to be different ?

p.7 Ass.3: $||\tau(X)||_\infty$ and $||c(X)||_\infty$ have not been defined.

p.7: the integrative $R$-learner has strictly larger efficiency -> what is meant here by efficiency ? This concept has not been properly defined. Do you mean that regularizing $c(X)$ imposes an implicit regularization on $\tau(X)$, which reduces the capacity of the RCT model, thus reducing model variance (at the cost of increasing model bias) ?

p.9: It is unclear to me what the "rct_x" and "os_x" methods are. The methods described in Section 3 and 5 seem to use both observational and experimental data. How are those methods modified to include only experimental (rct) or observational (os) data ?

p.9 Sec.6: the main claim of your paper is "without any prior knowledge of hidden confounding in the OS, the proposed integrative R-learner is consistent and asymptotically at least as efficient as the estimator using only the RCT". To support that claim experimentally, it is crucial that in your experiments, at least in the synthetic ones, you let the number of RCT samples vary from a small to a large number so as to verify that a) both estimators indeed converge to the same thing; and b) your estimator converges faster.

p.9: $U_{rct}$ has not been defined.

p.10 Tab.1: This experiment seems flawed. How is it that the unbiased RCT estimator "rct_rf" performs worse than the confounded one, "os_rf" ? It seems like the combination of RCT and OS data has a much lower impact on the final performance than the choice of the model family used to learn the estimator. I am not sure what "rct_oa", "os_oa", "rct_rlearner" and "os_rlearner" are, but if you want to evaluate the hypothesis that "int_rlearner" brings improvements due to the use of the OS data, then it is crucial that you also evaluate against a model that is learned from the RCT data only, which is adequate and properly regularized (e.g., using cross-validation).

p.10 Tab.1: Another comment, it seems crucial to me to include another important important baseline, which assumes unconfoundedness in the OS data, and estimates the RCT from a naive combination of RCT and OS data. I suspect that such a baseline, with a properly regularized model, could perform very well in this experiment.

p.10: generate population -> generate a population

p.10: slection -> typo

p.11 Tab.2: Here again, how is it that the biased estimators, using OS data only, perform better than the unbiased estimators, which use RCT data only ?

p.11: taking samples with $T=0$ -> what is $T$ here ?

p.11: an unbiased estimate from all samples -> $X$ is missing from this formula. How can this be an unbiased estimate of the RCT ?

p.12: the RMSE of our integrative R-learner increases -> how is it that the best RMSE obtained by your method is at iteration 1? Why event proposing an iterative learning procedure, if your method performs best without doing iterations ?


**Summary:**

This paper proposes a new method, integrative R-learner, for estimating causal effects from both observational and experimental data in the binary contextual bandit setting. Importantly, the observational data is subject to causal confounding, and both the observational and experimental data are subject to covariate shift (different distribution of the context variables). The authors show their method asymptotically converges to an unbiased estimator (under assumptions), at least as fast as the estimator that uses the experimental data.

---

> ### Author Response · Authors · 2021-12-04
> **Responses to Reviewer 8X2B part 1**
>
> We thank the reviewer for your careful reading of our manuscript and helpful comments. In what follows, we have the point-to-point responses to the detailed review comments.
> 1. Due to the confusions caused by the notations possibly varying in different communities, we first have clarifications for notations:
>     * The rule for the order of the operation we use: [{()}]. We start with (), then [], finally {}. For example, $2\times[1+2\times\\{1+2\times(1+1)\\}]=22$, we first calculate (...), then {...}, and finally  [...]. In eq2, A{...} means A$\times${...}.
>     *(0, 1) means the open interval from 0 to 1 without including 0 or 1, and [0, 1] means a closed interval including 0 and 1. We assume $P(A=1| X, S)\in(0, 1)$ instead of [0, 1] because the propensity score cannot take 0 or 1, which means that every subject has a positive probability to take either treatment.
>     *s=0, 1 means  $\forall s \in\\{0, 1\\}$.
>     *Y(a)⊥ S | X, (a=0, 1) means Y(a)⊥ S | X,  $\forall a \in\\{0, 1\\}$.
>     *$||\tau(X)||_\infty$ means $\sup_x\tau|(x)|$.
> 2. *“quite limited scope (binary contextual bandits).”*
>
>     As mentioned in the Discussion, “An interesting direction of future work is to extend the integrative R-learner to multiple treatments which commonly arise in reality. It can be constructed similarly based on the multivariate version of Robinson’s transformation and the corresponding R-learner (Section 7 in Nie and Wager, 2021)”.
> 3. *“many bells and whistles ... obfuscate the interpretability of the paper.”*
>
>     The proposed estimator is an easily implemented framework which can be summarized concisely as follows:
>     * Estimation of nuisance functions, i.e., P(A=1| X, S) and E(Y | X, S), with cross-fitting.
>     *Estimation of the HTE and confounding function with regularized regression.
>
>     Step a. can be achieved by many different models as long as they satisfy Assumption 4. We consider the series approximation with penalty and machine learning models (e.g., Xgboost) in simulation studies and real-data experiments. In Step b., we also consider both the series approximation with penalty and machine models (e.g., random forest). We use different models to illustrate that the proposed method is a general framework and can easily adopt suitable models in different situations. For a comparison of asymptotic variances of the integrative R-learner and the RCT-only estimator, we give the theoretical proof for the series approximation models.
> 4. *“e.g., the rct_x and os_x variants”*
>
>     The meanings of comparison methods are introduced in the beginning of Section 6. “rct_x” means using only the RCT alone to estimate the HTE with the method x. Similarly, “os_x” means using only the OS alone. For example, “rct_oa” stands for using only the RCT alone to estimate the HTE with the outcome adjusted method which is introduced in Section 5 (i.e., regressing the estimated adjusted outcome on X to estimate the HTE and confounding function).
> 5. *“p.3 Ass.1: I have a hard time understanding the meaning of (c).‘*
>
>     Assumption 1(c) is a causal consistency assumption which is a common assumption in causal inference literature (Rubin, 1980; Rosenbaum and Rubin, 1983). It means that an individual's potential outcome under his or her observed treatment is the outcome that will actually be observed for that person. This implies no interference: Potential outcomes for an individual are unaffected by treatments received or potential outcomes of other individuals; an exception is when the treatments are vaccines for prevention of an infectious disease.
> 6. *“p.6: the possible smooth or sparsity ... Do you still obtain gains with your method ?”*
>
>     We add regularization on both $\tau(x)$ and c(x). The efficiency gain is due to the relative complication of c(x) compared to $\tau(x)$. Intuitively, we can start with one extreme example where there is no unmeasured confounding, which means that the c(x)=0 (the simplest function). If we knew this fact, then we could combine the RCT and OS directly for HTE estimation, and the integrative estimator improves the efficiency of the RCT-only estimator obviously due to a larger sample size. Our proposed estimation method uses basis functions to approximate the confounding function and uses penalization to leverage the smoothness or sparsity of the confounding function to achieve the same result as the oracle one.

---

> ### Author Response · Authors · 2021-12-04
> **Responses to Reviewer 8X2B part 2**
>
> 7. *“To support that claim experimentally ... converges faster.”*
>
>     Thanks for the constructive comment. We added a simulation for the setting in Simulation I with $\tau(X)= 1+2X+0.75X^2, m=2000$ and the increasing n below, where “int_rlearner” converges to 0, which verifies the consistency.
> | n |rct_OA     |rwe_OA  |int_OA     |rct_rlearner |rwe_rlearner |int_rlearner |
> |:--|:----------|:-------|:----------|:------------|:------------|:------------|
> |200  |0.28(0.01) |0.57(0) |0.29(0.02) |0.26(0.01)   |0.58(0)      |0.23(0.01)   |
> |500 |0.16(0.01) |0.57(0) |0.13(0.01) |0.15(0.01)   |0.58(0)      |0.12(0.01)   |
> |1000 |0.11(0)    |0.57(0) |0.08(0)    |0.11(0)      |0.58(0)      |0.09(0.01)   |
> |2000 |0.08(0)    |0.57(0) |0.06(0)    |0.08(0)      |0.58(0)      |0.06(0)      |
> 8. *“p.9: $U_{rct}$  has not been defined.“*
>
>     $U_{rct}$ is an unobserved confounding variable which is generated from a normal distribution, please see Section 6.1.1.
> 9. *“p.11: taking samples with T -> what is here ?”*
>
>     This is a typo: it should be treatment A instead of T.
> 10. *“p.11: an unbiased estimate from all samples -> X is missing from this formula. How can this be an unbiased estimate of the RCT?”*
>
>     The propensity score in the equation is actually a special case of P(A=1 | X). We use a constant to estimate it because it is a randomized experiment. This is constructed following the method in Kallus et al. (2018).
> 11. *”p.12: the RMSE of our integrative R-learner increases ... without doing iterations ?”*
>
>     In the real-data experiment which has a much larger dimension of confounding variables than the simulated data because categorical variables are converted to dummy variables, machine learning approaches such as random forest have better performances than series approximation. However, unlike series approximation with explicit penalty terms, it is not straightforward to impose different regularization on $\tau(x)$ and c(x) with machine learning methods. Therefore, we propose an iterative method which estimates $\tau(x)$ and c(x) iteratively. The reason that the first iteration gives the best performance is that we use warm start when doing iterations, i.e., we initialize the confounding function values as the ones estimated from the method in Kallus et al. (2018) which is already a good approximation for the  c(x).
> 12. Thanks for pointing out the wording issues, other typos and providing two additional pieces of helpful literature. We have modified them in the revised version.

---

### Official Review · Reviewer_DtnT · 2021-11-25

**Confidence:** 3
**Overall Score:** 6

**Main Review:**

**Originality:**

The paper approaches a previously explored problem under less restrictive assumptions, with an estimator that generalizes a previously known one. This provides sufficient originality.

**Significance:**

The result that strict variance improvement over RCT data alone depends on the nature of basis functions for the effect and confounding functions is interesting—but not expanded upon.

It is slightly disappointing that advantage over RCT-only estimates is studied only in terms of asymptotic variance for a fixed proportion of RCT samples. If anything, this study is justified by the fact that RCT data is rare and observational data is plentiful. Although p(S=1) may be small, I find it hard to see that “n” grows linearly with “m” in practice. Admittedly, the comparison is made with an estimator that is more wasteful—which uses all N samples from an RCT. Nevertheless, it is unclear how the result scales with p(S=1) or for fixed “n” with growing “m”. Surely these questions are of greater practical concern in the settings used for motivation?

The comparison with Kallus et al. (2018) and Nie & Wager (2021) in section 5 is OK, it makes for a more self-contained paper, but does not add much new insight. For example, the instability of the U-learner is noted in existing work. Would it be possible to provide a rate comparison with the U-learner?

In simulation study I, the authors study the finite-sample properties of their estimator. This is a very low-dimensional setting (d=2) where the sample size from the RCT is proportionally quite large. Additionally, the model size is very small as well. The integrative r learner is the clear winner in this setting, yes, but it does not tell us much about where this edge comes from and if/when it is lost.

**Technical quality:**

Overlap between observational data and RCT is not discussed other than in experiments. However, is it not necessary for objective (5) to be uniquely identified in general? Specifically, if RCT and OS are completely disjoint, it would seem c and tau could be invariant to a mutliplication/division with a  constant on the observational data, since \tau does not have to be consistent with the RCT in these points.

The authors equate “If all the confounding variables are captured in the OS” with exchangeability w.r.t X. These are not equivalent, e.g., if the OS contains other post-treatment variables than the outcome.

In Assumption 1, \pi_A(X) is not defined.

**Clarity:**

The introduction is clear, but perhaps a little too long given its current content. This is not the first work studying a combination of RCT and OS data, and the problem does not need to be justified from scratch—the arguments that RCTs are expensive and OS data is plentiful but can lead to biased estimates have been made many times before. I appreciate that this gives a more self-contained description, but I think the space could be better spent. In particular, I’m missing references (in the introduction) to e.g., Bareinboim & Pearl (2016), Kallus, Puli & Shalit (2018)—especially since it is used heavily later in the paper, Peysakhovich & Lada (2016), Gentzel, Pruthi & Jensen (2021), etc. While these may not address the same exact setting as is studied here, they are more closely relevant than some of the other references in the introduction. In particular, this could help contextualize the contributions of the present work.

The Setup section clearly defines the problem and the main assumptions.
Section 3.1 justifies the proposed learning objective well.

Unless I’m mistaken, Algorithm 1 describes a special case of grid search. I’m not sure an algorithm box is necessary here—although it is certainly clear!

I think the abbreviations NK_linear and NK_lasso could be interpreted as “Nathan Kallus_linear”. Surely, it would be more appropriate to also give credit to the two co-authors of that paper than to list both first and last name of the first author?

The original R-learner studies a non-parametric kernel-based estimator. Here, a series approximation is used. Was there a particular reason for this departure?


**Summary:**

The paper studies estimation of causal effects using a combination of experimental and observational data. The authors propose the integrative R-learner, an extension of the existing R-learner estimator, for the purpose. Under the assumptions of effect transportability and ignorability in the experimental data, this is shown to lead to consistent estimation with asymptotic variance at least as good as a comparable estimator based only on experimental data. The approach is evaluated on synthetic and real-world data and compared with relevant baselines.

---

> ### Author Response · Authors · 2021-12-04
> **Responses to Reviewer DtnT**
>
> We thank the reviewer for your careful reading of our manuscript and helpful comments. In what follows, we have the point-to-point responses to the detailed review comments.
> 1. *“The result that strict variance improvement over RCT data alone ... but not expanded upon.”*
>     Thanks for the constructive comments. To have a better understanding of strict variance improvement over the RCT-only estimator, we consider an extreme example where the OS does not have unmeasured confounding. In this example, the confounding function is zero. If we knew this fact priori, we can combine the experimental and observational studies directly for HTE estimation. The integrative R-learner improves the efficiency of the RCT-only estimator for the obvious reason that it uses a larger sample size. As long as the confounding function has a simpler structure than the HTE, e.g. more sparse in variables or more smooth in its functional form, the integrative method outperforms the RCT-only estimator. When we lack such prior knowledge for the confounding function, we use basis functions to approximate the confounding function and use penalization to exclude the redundant basis functions and leverage the sparse or smooth structure of the confounding function to achieve the efficiency gain as the oracle one does.
> 2. *“It is slightly disappointing that advantage over RCT-only estimates is studied only in terms of asymptotic variance, … in the settings used for motivation?”*
>     Thanks for the inspiring feedback. Our proposed method is indeed motivated in the settings where the sample size of the RCT (n) is small but the sample size of the OS (m) is much larger. To address the practical concern, we can allow n/m go to 0 with n and m going to infinity in our asymptotic regime. This asymptotic regime is not restrictive in some practical situations when the RCT has hundreds of patients and the OS has a much larger sample. We agree that non-asymptotic risk bounds for a fixed n and a growing m are also of great interest, and we will work on deriving such bounds in our future work.
> 3. *“Would it be possible to provide a rate comparison with the U-learner?”*
>     Thanks for the great comment. Yes, we can also show that the U-learner is root-N consistent for the HTE as our proposed integrative R-learner. The proof steps are similar: using sample splitting to estimate the nuisance functions helps to replace the estimated nuisance functions in the adjusted outcomes in the U-learner with the true ones, and then applying the theory of series approximation leads to the asymptotic properties of the U-learner. Although the U-learner and the R-learner have a comparable rate of convergence, we chose to extend the R-learner to the data integration context exactly because the R-learner is more stable than the U-learner, especially when the propensity score can be close to zero or one.
> 4. *“Additionally, the model size is very small … where this edge comes from and if/when it is lost.”*
>     The efficiency gain comes from the smoothness or sparsity of the confounding function (Theorem 1). When the confounding function requires more basis expansion than the HTE, then the integrative approach does not gain any efficiency on HTE estimation. We agree that additional simulation studies will be needed to provide a comprehensive evaluation of the proposed method.
> 5. *“Overlap between observational data and RCT is not discussed other than in experiments.”*
>     Thanks for bringing up this important question. The goal is to identify and estimate (x), where x belongs to the support of X in RCT. The support of X in RCT must overlap with/nest in the support of X in OS. Outside of the support of X in RCT, (x) is not identifiable. We will add this assumption in the revised paper.
> 6. *“The authors equate “If all the confounding variables are captured in the OS” ... other post-treatment variables than the outcome.”*
>     Thanks for the comment. We will clarify this as “If all the pre-treatment confounding variables are captured in the OS”.
> 7. *“In Assumption 1, \pi_A(X) is not defined.”*
>     This is a typo: \pi_A(X) should be changed to e(X, S). We will make the correction.
> 8. *“Unless I’m mistaken, Algorithm 1 describes a special case of grid search. I’m not sure an algorithm box is necessary here—although it is certainly clear!”*
>      We agree with the comment, and we will move Algorithm 1 to the appendix if the space is limited.
> We will change “NK_linear” and “NK_lasso” to “KPS_linear” and “KPS_lasso” so as to include the first letter of the last names of all the authors.
> 9. *“The original R-learner studies a non-parametric kernel-based estimator. Here, a series approximation is used. Was there a particular reason for this departure?”*
>     Using series approximations enables us to compare the efficiency gain in Theorem 1 more easily.
> 10. Thanks for the suggestion on the additional related literature in the introduction part. We will add them in the revised paper.

---

> > ### Comment · Reviewer_DtnT · 2021-12-25
> > **Re: response**
> >
> > Thank you for your response!

---

### Decision · Program_Chairs · 2022-01-12

**Decision:**

Accept (Poster)

**Comment:**

The paper studies an extension of the R-learner to settings where both experimental and observational data are available. The author show that asymptotically the proposed estimator converges to the truth at least as fast as the estimator that uses only experimental data. This is an interesting extension of important work and thus well-suited for CLEAR. However, as some reviewers point out, the paper is hard to read due to use of dense notation and there are some issues with the simulations. It would be great if in the final revision the authors could make an attempt at improving readability, for example by addressing the comments by reviewer 8X2B.